# Myogenin is an essential regulator of adult myofibre growth and muscle stem cell homeostasis

**Massimo Ganassi[1]\*, Sara Badodi[2], Kees Wanders[1], Peter S Zammit[1], Simon M Hughes[1]\***

[1]Randall Centre for Cell and Molecular Biophysics, King's College London, London, United Kingdom; [2]Blizard Institute, Barts and The London School of Medicine and Dentistry, Queen Mary University of London, London, United Kingdom

**Abstract** Growth and maintenance of skeletal muscle fibres depend on coordinated activation and return to quiescence of resident muscle stem cells (MuSCs). The transcription factor Myogenin (Myog) regulates myocyte fusion during development, but its role in adult myogenesis remains unclear. In contrast to mice, $myog^{-/-}$ zebrafish are viable, but have hypotrophic muscles. By isolating adult myofibres with associated MuSCs, we found that $myog^{-/-}$ myofibres have severely reduced nuclear number, but increased myonuclear domain size. Expression of fusogenic genes is decreased, Pax7 upregulated, MuSCs are fivefold more numerous and mis-positioned throughout the length of $myog^{-/-}$ myofibres instead of localising at myofibre ends as in wild-type. Loss of Myog dysregulates mTORC1 signalling, resulting in an 'alerted' state of MuSCs, which display precocious activation and faster cell cycle entry ex vivo, concomitant with $myod$ upregulation. Thus, beyond controlling myocyte fusion, Myog influences the MuSC:niche relationship, demonstrating a multi-level contribution to muscle homeostasis throughout life.

**\*For correspondence:**
massimo.ganassi@kcl.ac.uk (MG);
simon.hughes@kcl.ac.uk (SMH)

**Competing interests:** The authors declare that no competing interests exist.

## Introduction

Maintenance of adult skeletal muscle depends on the ability of multinucleated myofibres to grow and regenerate, thereby ensuring optimal functionality throughout life. To facilitate this homeostasis, adult vertebrate muscle contains a specialised population of precursors cells, muscle stem cells (MuSCs), also termed satellite cells, located between the basal lamina and sarcolemma of most adult myofibres (*Mauro, 1961*). Like other stem cells, MuSCs are mitotically dormant in normal circumstances but poised to respond to functional demand for new myonuclei throughout adult life (*Relaix and Zammit, 2012*; *Purohit and Dhawan, 2019*). The transcription factor Pax7 is considered a canonical quiescent MuSC marker across several vertebrate species and its expression is maintained during the progression to activation and proliferation, before downregulation at the onset of myogenic differentiation (*Berberoglu et al., 2017*; *Buckingham and Relaix, 2015*; *Chen et al., 2006*; *Hammond et al., 2007*; *Hollway et al., 2007*; *Kawakami et al., 1997*; *Seale et al., 2000*; *Seger et al., 2011*; *Zammit et al., 2006*; *Olguin and Olwin, 2004*). Null mutations for Pax7 severely affects MuSC maintenance and muscle regeneration in amniotes, amphibia and teleosts (*Berberoglu et al., 2017*; *Chen et al., 2006*; *Oustanina et al., 2004*; *Relaix et al., 2006*; *Seale et al., 2000*). In adult zebrafish, Pax7[+] MuSCs contribute to regeneration of myofibres upon muscle damage, being the functional counterparts of MuSCs in mammals (*Berberoglu et al., 2017*; *Hollway et al., 2007*; *Pipalia et al., 2016*).

Proper function of the MuSC pool depends on a dynamic balance between quiescence and activation and is sustained by feedback signalling from surrounding muscle (*Mashinchian et al., 2018*; *Forcina et al., 2019*). In response to stimuli, quiescent MuSCs activate and become muscle

progenitor cells (MPCs), which proliferate, differentiate and fuse to contribute myonuclei to pre-existing multinucleated myofibres: a process that resembles aspects of myogenesis by embryonic myoblasts and relies on an intricate molecular network comprising both extrinsic and intrinsic mechanisms (*Buckingham and Relaix, 2015*). Among many converging factors, the members of the Myogenic Regulatory Factor (MRF) family of transcription factors, Myod, Myf5, Mrf4 and Myogenin (Myog), are key regulators of vertebrate muscle gene expression during both early and adult myogenesis (*Zammit, 2017*; *Hernández-Hernández et al., 2017*). In adult muscle, Myod and Myf5 are mainly expressed in MuSC and are crucial for efficient activation and proliferation (*Cooper et al., 1999*; *Kuang et al., 2007*; *Megeney et al., 1996*; *Soleimani et al., 2012*), whereas Mrf4 accumulates in mature myofibres and contributes to ensure their homeostatic size regulation (*Moretti et al., 2016*; *Voytik et al., 1993*).

Knockout of the *Myog* gene in mouse leads to severe muscle deficiencies and, in contrast to the other MRFs, is neonatal lethal (*Hasty et al., 1993*; *Nabeshima et al., 1993*), thus making it difficult to investigate Myog function in detail. As in embryogenesis, *Myog* expression is upregulated in proliferating MuSCs that then rapidly undergo myogenic differentiation (*Zammit et al., 2004*) and its expression appears reciprocally balanced with that of *Pax7* (*Olguin et al., 2007*; *Riuzzi et al., 2014*; *Zammit et al., 2006*; *Olguin and Olwin, 2004*). However, depletion of mouse Myog in myoblasts does not block accumulation of differentiation markers in vitro, but cell biological aspects such as myotube formation were not explored (*Meadows et al., 2008*). We recently expanded this observation, reporting that, despite being dispensable for myogenic differentiation, Myog is essential for most myocyte fusion and its functional depletion leads to formation of mononucleated myofibres and reduced myotome growth during zebrafish embryonic/larval stages (*Ganassi et al., 2018*). Moreover, modulation of rodent Myog expression shifts muscle enzyme activity towards oxidative metabolism, alters exercise capacity and is required for neurogenic atrophy (*Flynn et al., 2010*; *Meadows et al., 2008*; *Moresi et al., 2010*; *Ekmark et al., 2003*; *Hughes et al., 1999*), suggesting a broad Myog role in regulating adult muscle homeostasis. Nevertheless, whether Myog acts on myofibre, MuSC or both has remained elusive. Importantly, in contrast to mouse, zebrafish *myog*[-/-] mutant adult fish are alive, albeit displaying hypotrophic muscle (*Ganassi et al., 2018*). Congruent with the fish phenotype, almost complete depletion of mouse *Myog* after birth results in decreased muscle size (*Knapp et al., 2006*; *Meadows et al., 2008*), thereby confirming a conserved vertebrate role for Myog in regulating bulk muscle growth and maintenance.

Here, to explore the role of Myog in adult muscle, we analysed *myog*[-/-] mutant zebrafish and deployed our recently developed method to isolate viable single myofibres with associated MuSCs, allowing culture of muscle progenitor cells (MPCs). Morphometric analysis showed that *myog*[-/-] muscle has reduced myofibre growth and nuclear accretion compared to siblings (sibs), revealing increased myonuclear domain size. Adult *myog*[-/-] muscle exhibited upregulation of *pax7a* and *pax7b* expression along with supernumerary MuSCs, that were randomly distributed throughout myofibre length compared to sib MuSCs, which localise mainly at myofibre ends, suggesting an altered MuSC niche. Ex vivo analysis of *myog*[-/-] MuSCs revealed elevated phosphorylation of ribosomal protein S6, concomitant with reduced expression of the mTOR inhibitors *tsc1a* and *tsc1b*, indicating enhanced mTORC1 activity, which marks MuSCs that have left deep quiescence to enter into an 'alert' pseudo-activated state (*Rodgers et al., 2014*). Congruently, cultured myofibres from mutants yielded increased numbers of MPCs with an accelerated proliferative phase, marked by upregulated expression of *pax7a*, *pax7b*, and *myod*. Together, our study demonstrates that Myog plays a crucial role in adult muscle growth and MuSC homeostasis.

## Results

### Adult *myog*[-/-] myofibres are small with fewer nuclei but increased nuclear domain size

Adult zebrafish bearing the *myog*[kg125] nonsense mutant allele (*myog*[-/-]), which reduces *myog* mRNA, eliminates Myog protein and is presumed null, are viable but have reduced muscle bulk (*Ganassi et al., 2018*). In contrast, *myog*[kg125/+] heterozygous siblings (sib) are indistinguishable from wild-type sibs and are used as paired controls throughout the current study. Gene expression analysis on dissected adult trunk muscle confirmed continued significant downregulation of *myog* mRNA

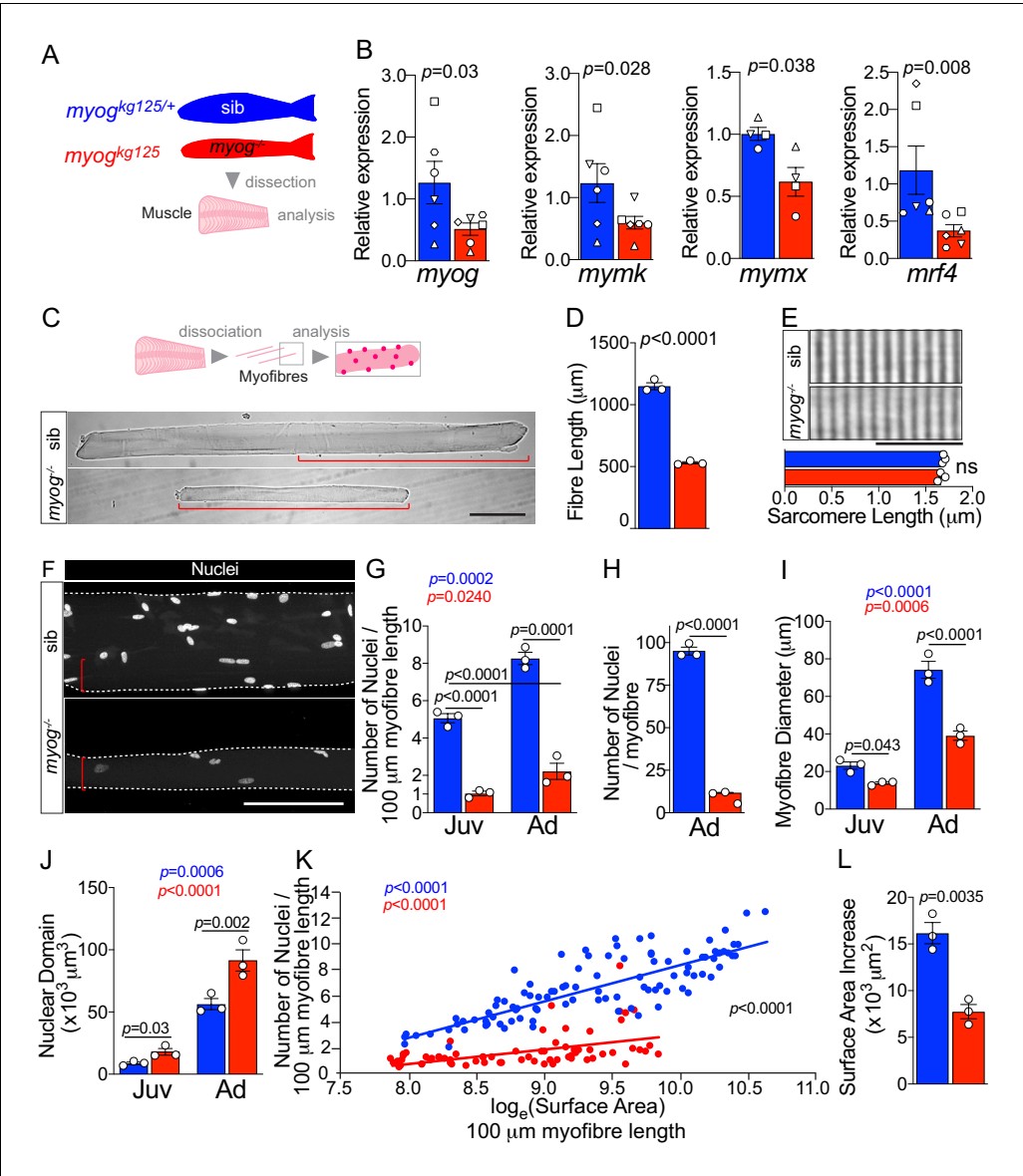

**Figure 1.** Myogenin is required for normal myofibre size and nuclear accretion. (**A**) Schematic of trunk muscle processing for analysis, pink represents fish fillet. Colours identify sib (*myog*$^{kg125/+}$, blue) or *myog*$^{-/-}$ (*myog*$^{kg125}$, red) samples throughout the figure. (**B**) qPCR analysis shows downregulation of *myog*, *mymk*, *mymx* and *mrf4* mRNAs in adult *myog*$^{-/-}$. Symbol shapes denote paired sib and *myog*$^{-/-}$ samples, n = 4–6 fish/genotype, paired *t*-test. (**C**) Schematic of myofibre isolation for morphometric analysis (top) and representative images (bottom) showing smaller *myog*$^{-/-}$ myofibre (red brackets) compared to age-matched sib. Scale bar = 100 μm. (**D**) Measure of absolute myofibre length, n = 3 fish/genotype, n = 110–120 myofibres/fish, unpaired *t*-test. (**E**) Representative images and measure of unaltered sarcomere length on freshly isolated myofibres, n = 3 fish/genotype, n = 10 myofibres/fish, unpaired *t*-test. Scale bar = 10 μm. (**F**) Representative images of isolated fixed adult myofibres show size reduction in *myog*$^{-/-}$ (red brackets). Scale bar = 100 μm. (**G–J**) Quantification of number of nuclei/100 μm (**G**), absolute number of nuclei per myofibre (**H**), myofibre diameter (**I**), and nuclear domain size (myofibre volume per nucleus) (**J**) showing significant changes in myofibres from juvenile (Juv, 1 month-old) and adult (Ad, 8 months-old) stages within (coloured *p*) or among (black *p*) genotypes. n = 3 fish/genotype, n = 30–50 adult myofibres/fish, n = 15–20 juvenile myofibres/fish, one-way ANOVA. (**K**) Relationship of number of nuclei and log$_e$(Surface Area) indicates different growth mode between sib and *myog*$^{-/-}$ (i.e. significant slope difference, black *p*), despite significant correlation of log$_e$(SA) with nuclear number within genotype (coloured *p*) (see Materials and methods). (**L**) Increase in Surface Area (SA) from juvenile to adult stage (=Ad_SA – Juv_SA) indicates reduced growth rate in *myog*$^{-/-}$. Data from ***Figure 1—figure supplement 1E***, unpaired *t*-test. All graphs report mean ± SEM. Statistical significance within (coloured *p*) or between (black *p*) genotypes is indicated.

*Figure 1 continued on next page*

*Figure 1 continued*

The online version of this article includes the following figure supplement(s) for figure 1:

**Figure supplement 1.** *Myog^kg125* mutant adult and juvenile myofibres are smaller.

in *myog^-/-* fish compared to co-reared sib controls (*Figure 1A,B*) in line with the nonsense-mediated decay reported previously (*Ganassi et al., 2018*). Levels of mRNA encoded by the terminal myogenesis genes *mymk (tmem8c)*, *mymx* and *mrf4 (myf6)* were downregulated by 52%, 38% and 68%, respectively, in *myog^-/-* compare to the relative expression in sib, as during the embryonic stage, suggesting a continuous muscle defect in adulthood (*Figure 1B*). To explore further the muscle defect at a cellular level, we isolated single myofibres from juvenile (1 mpf; months post-fertilisation) and adult (8 mpf) *myog^-/-* and sibs (*Figure 1C*). Both juvenile and adult mutant myofibres were on average 20% and 50% shorter than those in control, respectively, demonstrating an early onset growth deficit in *myog^-/-*, with lifecourse worsening (*Figure 1C,D* and *Figure 1—figure supplement 1A,B*). Despite significant 50% reduction in overall fibre length, sarcomere length was unaffected in adult *myog^-/-* myofibres, as in embryonic myofibres (*Figure 1E*; *Ganassi et al., 2018*). In contrast, myofibres isolated from the adult hypomorphic *myog^fh265* mutants (*Hinits et al., 2011*; *Ganassi et al., 2018*) were indistinguishable from those of age-matched sibs (*Figure 1—figure supplement 1C*). The number of nuclei associated with each isolated myofibre appeared greatly reduced in *myog^-/-* mutants compared to sibs (*Figure 1F* and *Figure 1—figure supplement 1D*). Indeed, nuclear numbers per unit length were reduced fourfold in mutants (*Figure 1G*). Congruently, calculation of the average number of nuclei associated with each adult myofibre by multiplying the number of nuclei/unit length by the length revealed an approximate 90% reduction, from an average of 95 nuclei per myofibre in sibs to nearly 12 nuclei per myofibre in *myog^-/-* (*Figure 1H*). Number of total nuclei in *myog^-/-* myofibres was reduced similarly at juvenile stage compared to sibs (*Figure 1—figure supplement 1D*). Juvenile sib myofibres had significantly more nuclei than adult *myog^-/-* myofibres, confirming ongoing myonuclear accretion defect in *myog^-/-* (*Figure 1G,H* and *Figure 1—figure supplement 1D*). Thus, *myog^-/-* mutant myofibres are grossly defective and show no sign of homeostatic recovery over time.

Further morphometric analysis revealed that lack of Myog led to an average persistent decrease of 40% in myofibre diameter resulting in 50% reduction of myofibre surface area, compared to age-matched juvenile and adult sib, respectively (*Figure 1F,I* and *Figure 1—figure supplement 1E*). Indeed, as expected, both the volume and surface area per fibre-associated nucleus (nuclear domain and surface area domain size (SADS), respectively) were increased by almost twofold and threefold, respectively, in *myog^-/-* (*Figure 1J* and *Figure 1—figure supplement 1F*) indicating both inadequate nuclear accretion, congruent with suppression of *mymk* and *mymx* mRNAs (*Figure 1B*), and a compensatory increase in nuclear domain and SADS, which remained larger in mutants throughout life, showing no sign of return to a normal size with time. Nevertheless, both genotypes displayed significant increase in both myofibre diameter, surface area (SA) and number of nuclei between juvenile and adult stages, indicating ongoing muscle growth (*Figure 1G,I* and *Figure 1—figure supplement 1E*). However, although myofibre SA and number of nuclei positively correlated in *myog^-/-*, we observed a reduced nuclear accretion as myofibres acquired larger area compared to sib, hence highlighting a significantly different growth mode between the two genotypes (different trend slopes) (*Figure 1K*). Coherently, the incremental increase in myofibre SA over the juvenile-to-adult time-frame was significantly reduced in *myog^-/-*, confirming a persistent deficit in muscle growth (*Figure 1L*). Thus, loss of Myog function affects normal muscle growth, impinging on size and nuclear accretion of myofibres.

## Adult *myog^-/-* myofibres have increased number of MuSCs

Isolated myofibres have two kinds of associated nuclei, genuine muscle fibre nuclei and nuclei of resident MuSCs. To assess MuSC abundance in *myog^-/-*, we first analysed *pax3* and *pax7* mRNAs, well-known markers of MuSCs (*Buckingham and Relaix, 2015*; *Pipalia et al., 2016*; *Hammond et al., 2007*). Gene expression analysis on whole muscle showed upregulation of *pax7a* and *pax7b* mRNAs of 82% and 120%, respectively, in *myog^-/-* compared to control sibs, but no change in *pax3a* or *pax3b* (*Figure 2A* and *Figure 2—figure supplement 1A*). *Myf5* mRNA, a marker of partially-

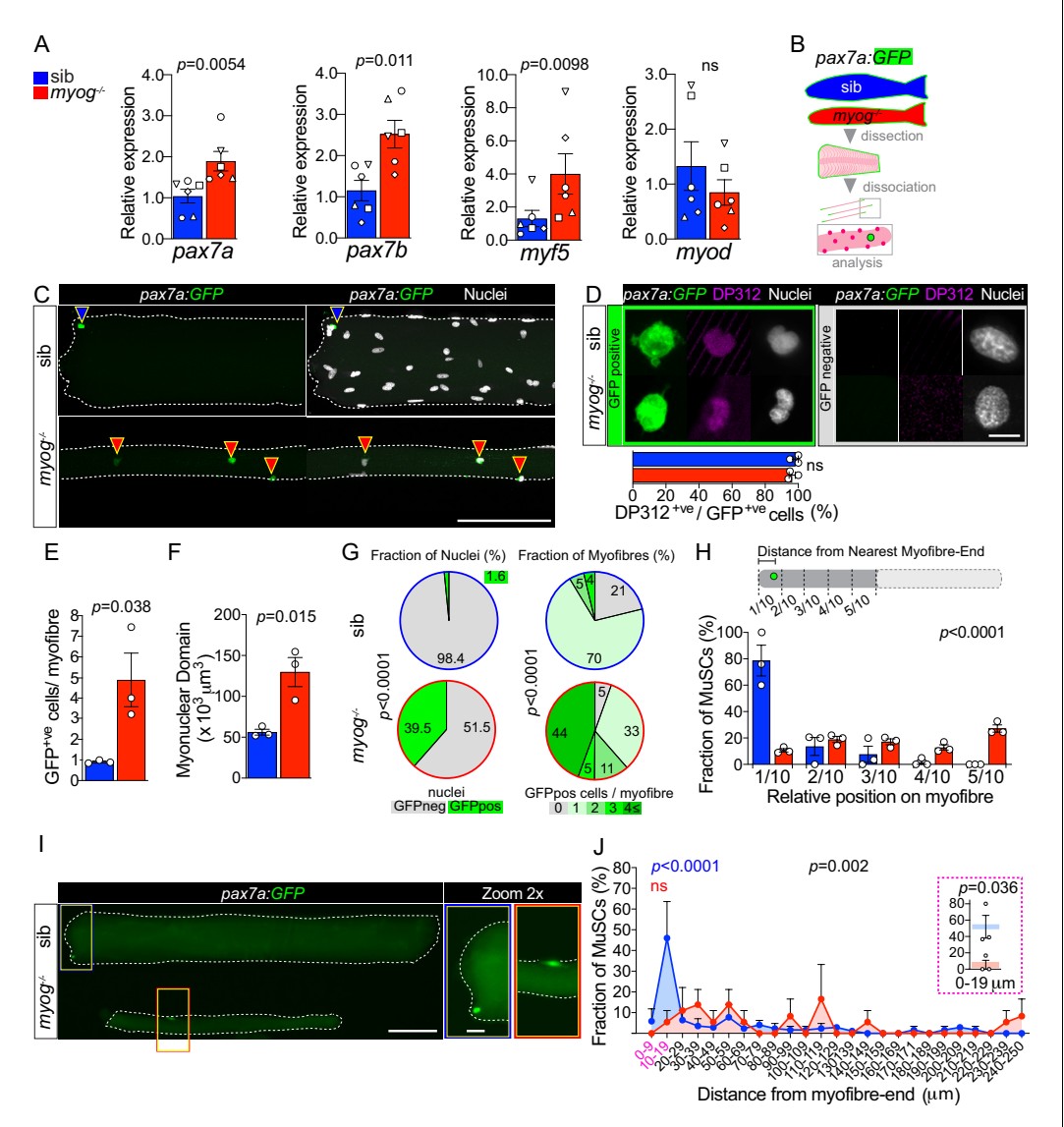

**Figure 2.** Lack of Myogenin alters MuSC number and localisation. Colours identify sib (blue) or *myog*$^{-/-}$ (red) samples throughout the figure. (**A**) qPCR analysis showing upregulation of *pax7a*, *pax7b*, *myf5* and unaltered *myod* in adult *myog*$^{-/-}$ mutant muscle. Symbol shapes denote paired sib and *myog*$^{-/-}$ samples, n = 6 fish/genotype, paired *t*-tests. (**B**) Schematic of myofibre isolation for GFP$^{+ve}$ cells analysis from *pax7a:GFP;myog*$^{-/-}$ and sib muscles. (**C**) Representative images showing GFP$^{+ve}$ cells on sib (blue arrowhead) or *myog*$^{-/-}$ (red arrowheads) on isolated myofibres. Scale bar = 100 µm. (**D**) Representative immunodetection of GFP (green), DP312 (Pax3/Pax7; magenta) and nuclei (white) on GFP$^{+ve}$ (positive, green panel) cells or GFP$^{-ve}$ (negative, grey panel) nuclei on freshly isolated adult sib or *myog*$^{-/-}$ myofibres. Scale bar = 5 µm. Quantification of DP312 immunostaining on *pax7a:GFP* MuSCs confirms the stem cell identity of both sib (n = 36/genotype) and *myog*$^{-/-}$ (n = 67/genotype) GFP$^{+ve}$ cells, n = 3 fish/genotype unpaired *t*-test. (**E,F**) Quantification of absolute number of GFP$^{+ve}$ cells (*bona fide* MuSC) per myofibre and myonuclear domain size. n = 3 fish/genotype, n = 20–30 myofibres/fish, unpaired *t*-test. (**G**) Pie charts showing the fraction of nuclei (left) and fraction of myofibres (right) with the indicated number of nuclei in GFP$^{+ve}$ cells (left) or cells/myofibre (right) in sib (blue circles) and *myog*$^{-/-}$ (red circles). Data from *Figure 2—figure supplement 1G*. *p*-values indicate probability of rejecting null hypothesis of no difference between *myog*$^{-/-}$ and sib in $\chi^2$ tests. (**H**) Diagram of measurement of GFP$^{+ve}$ cell distance from nearest myofibre-end (top). Each myofibre was segmented into tenths, where 1/10 and 5/10 corresponded to the segments nearest to and furthest from the myofibre-end, respectively. Quantification of fraction of MuSC located within each myofibre segment (bottom, see Materials and methods and *Figure 2—figure supplement 1H*). n = 3 fish/genotype, n = 30–50 MuSCs, *p*-value indicates probability of rejecting null hypothesis of no difference between *myog*$^{-/-}$ and sib in $\chi^2$ test. (**I**) Representative images showing localisation of the GFP$^{+ve}$ cell on sib (blue rectangle) or *myog*$^{-/-}$ (red rectangle) mono-MuSC myofibres (with only one GFP$^{+ve}$ cell). Scale bar = 100 µm (=20 µm in 2x-zoom). (**J**) Quantification of fraction of MuSC located within each 10 µm segment from mono-MuSC myofibre-end. n = 3 fish/genotype, n = 15–25 MuSCs, black *p*-value indicates probability of rejecting null hypothesis of no difference between *myog*$^{-/-}$ and sib in $\chi^2$ test. Coloured *p*-values indicate one-way ANOVA analysis of non-random MuSC distribution in 10 µm

*Figure 2 continued on next page*

*Figure 2 continued*

segments within each genotype. Magenta inset reports fraction of sib and *myog*[-/-] MuSC within the 0–9 and 10–19 µm segments, unpaired *t*-test. Symbols show results for individual fish. All graphs report mean ± SEM.

The online version of this article includes the following figure supplement(s) for figure 2:

**Figure supplement 1.** *pax7a:GFP;myog*[kg125] muscle is smaller and displays altered MuSC cellularity and unchanged *Pax3* genes expression.

activated or 'alerted' MuSCs in mice and MPCs in fish, and known to be a Pax7 target gene (*Coutelle et al., 2001*; *Kuang et al., 2007*; *Soleimani et al., 2012*), was upregulated by over 200%, further suggesting an increase of MuSCs number in *myog*[-/-]. In contrast, *myod* mRNA, a marker of fully-activated proliferating MuSCs (*Zammit et al., 2004*; *Zammit et al., 2002*; *Megeney et al., 1996*), was unaltered (*Figure 2A*). These data raise the possibility that *myog*[-/-] adults have alterations in MuSCs. To facilitate identification of *bona fide* MuSCs, *myog*[kg125/+] fish were bred onto *pax7a:GFP* reporter and MuSCs identified by GFP fluorescence (*Figure 2B,C*; *Pipalia et al., 2016*; *Mahalwar et al., 2014*). Immunostaining of GFP-positive (GFP[+ve]) cells with the DP312 antibody, that recognises fish Pax3/Pax7 (*Hammond et al., 2007*; *Davis et al., 2005*), revealed co-labelling of virtually all GFP[+ve] cells, thus confirming MuSC identity (*Figure 2D*). Adult *pax7a:GFP;myog*[-/-] fish showed reduced muscle compared to co-reared *pax7a:GFP;myog*[+/-] sibs (*Figure 2—figure supplement 1B*), replicating the phenotype on a wild-type background (*Ganassi et al., 2018*). Upon muscle dissociation, we observed that single *myog*[-/-] myofibres bore around five GFP[+ve] cells on average, compared to only one on sib myofibres (*Figure 2E*). As we previously showed that cytoplasmic GFP diffuses throughout the cytoplasm of muscle fibres within seconds after fusion leading to a dramatic weakening of signal (*Pipalia et al., 2016*; *Bajanca et al., 2015*), these data show that MuSCs are more numerous on mutant myofibres. Indeed, almost 40% of nuclei associated with *myog*[-/-] myofibres were in GFP[+ve] cells compared to less than 2% in sib (*Figure 2G* and *Figure 2—figure supplement 1C*). Thus, MuSCs are more abundant in *myog*[-/-] mutants than in sibs. Subtraction of MuSC nuclei from the total nuclei associated with isolated adult myofibres, revealed that sib myofibres have over thirteenfold more genuine myofibre nuclei (i.e. myonuclei) than *myog*[-/-] myofibres (*Figure 2—figure supplement 1D*). These GFP-negative (GFP[-ve]) myonuclei allowed calculation of a myofibre volume per true myonucleus, the notional myonuclear domain. In line with SADS, the myonuclear domain was increased by 2.3-fold in *myog*[-/-] fibres (*Figure 2F*), despite the eightfold reduction in absolute myofibre volume (*Figure 2—figure supplement 1E*). Accordingly, corrected SADS showed an increase of almost fivefold in mutant (*Figure 2—figure supplement 1F*). Thus, despite *myog*[-/-] fish having less muscle due to reduced fusion, Myog in the adult also regulates muscle growth by limiting both nuclear domain size and MuSC number.

## Altered MuSC:niche relation in *myog*[-/-] mutant myofibres

Myofibres from *myog*[-/-] showed a striking set of alterations in MuSC distribution. In control sib, MuSCs were found mostly localised close to the myofibre end, whereas in *myog*[-/-] the supernumerary MuSCs were randomly distributed along the myofibre length (*Figure 2H* and *Figure 2—figure supplement 1G*). In sibs, 70% of myofibres had a single associated MuSC, which was usually within 20 µm from the myofibre end (*Figure 2G,I,J* and *Figure 2—figure supplement 1G,H*). In contrast, only 33% of myofibres had a single MuSC in mutants (*Figure 2G,I* and *Figure 2—figure supplement 1G,H*), which displayed consistent random location, often far from the myofibre end (*Figure 2I,J*). Around 60% of *myog*[-/-] myofibres bore two or more MuSCs, whereas only 9% of sib myofibres had two or more MuSCs, and none had over three (*Figure 2G*). Strikingly, 21% of sib myofibres had no GFP[+ve] cells, compared to only 5% of *myog*[-/-] myofibres (*Figure 2G* and *Figure 2—figure supplement 1G*). The results suggest that lack of Myog either enhances MuSC proliferation, prevents MuSC differentiation or both. Data in this and the preceding section demonstrate that lack of Myog alters MuSC number, position and expression of stemness marker genes, suggestive of an alteration in their relationship to their normal myofibre end niche.

## Lack of Myog enhances mTORC1 signalling in adult muscle and MuSCs

To investigate further how Myog may influence the MuSC niche, we assessed the expression of factors that contribute to muscle growth via regulation of MuSC activation. Whole muscle analysis

revealed 130% increased level of *igf1* mRNA and nearly 50% downregulation of *myostatin b* (*mstnb*) mRNA, both regulators of muscle growth and the mTORC1 (mechanistic Target Of Rapamycin Complex 1) pathway (*Yoon, 2017*; *Trendelenburg et al., 2009*), suggesting that lack of Myog triggers a growth signalling response (*Figure 3A,B*). In contrast, expression of the zebrafish IGF1 receptors, *igfr1a* and *igfr1b* was unaltered (*Figure 3—figure supplement 1*) indicating that *myog*[-/-] myofibres, and/or MuSCs, may respond to the higher IGF1 level similarly to sib. IGF1 signalling involves phosphorylation of AKT that activates mTORC1 resulting in downstream events such as promotion of muscle growth, accompanied by phosphorylation of the ribosomal protein S6 (RPS6) (*Figure 3C* and *Yoon, 2017*; *Ruvinsky et al., 2009*). Western blot analysis of whole muscle extracts revealed that whereas AKT phosphorylation at Ser473 (pAKT) appeared unchanged, downstream phosphorylation of RPS6 at Ser240/244 (pRPS6) was enhanced fourfold in *myog*[-/-] muscle compared to sib (*Figure 3D*). Phosphorylation of RPS6 (pRPS6) marks the onset of MuSC activation in vivo (*Rodgers et al., 2014*). Thus, we analysed pRPS6 level in MuSCs on freshly isolated myofibres from *pax7a:GFP;myog*[-/-] or control *pax7a:GFP;myog*[+/-] sib. *Myog*[-/-] MuSCs showed a robust twofold increase in pRPS6 level (*Figure 3E,F*), in line with quantification in whole muscle. No change was detected in myofibres themselves. As pRPS6 was increased but pAKT was not similarly altered, we reasoned that lack of Myog might control the expression level of factors that inhibit the mTORC1 pathway. Tsc1 and Tsc2 form a stabilised GTPase complex that represses the mTOR cascade (*Figure 3C*; *Nobukini and Thomas, 2004*). Levels of *tsc1a* and *tsc1b* mRNAs were reduced by 51% and 57%, respectively, in *myog*[-/-] muscle, whereas *tsc2* mRNA was unchanged compare to sib (*Figure 3G*), consistent with increased pRPS6. We conclude that lack of Myog triggers activation of the mTORC1 pathway in MuSCs, concomitant with *tsc1a/b* repression, leading to precocious MuSC activation.

## Adult *myog*[-/-] MuSC-derived myoblasts exhibit faster transition to proliferation

To assess MuSC activation and proliferation potential, myofibres bearing GFP[+ve] cells, from either *pax7a:GFP;myog*[-/-] or control *pax7a:GFP;myog*[+/-] sib were individually plated, cultured for 4 days in growth medium and the number of associated myogenic cells counted (*Figure 4—figure supplement 1A*). *Myog*[-/-] mutants yielded at least three times as many mononucleate cells than did sibs (*Figure 4—figure supplement 1A*), paralleling the increased number of MuSCs on freshly-isolated myofibres. The vast majority (80–90%) of cells from fish of either genotype was GFP[+ve], so *bona fide* MPCs. *Myog*[-/-] myofibres also displayed an average threefold increased GFP[+ve] cell yield compared to sibs (*Figure 4A–C*). Within the 4-day culture period, sib myofibres (70% of which had only a single GFP[+ve] MuSC) yielded around 40 MPCs, reflecting a mean doubling time of around 20 hr. Mutants yielded a slightly higher proportion of GFP[+ve] MPCs than sibs (*Figure 4C*), perhaps reflecting the increased number of MuSCs on each myofibre and the upregulation of *pax7a* mRNA in *myog*[-/-] muscle. Importantly, although both genotypes yielded a GFP[-ve] cell population, all cells immunoreacted for Desmin, confirming their myogenic identity (*Figure 4—figure supplement 1A*). Notably, the relative MPC proliferation rates were not significantly different among genotypes, calculated as the ratio of the number of GFP[+ve] MPCs obtained after 4 days in culture to the average number of GFP[+ve] MuSC per myofibre at the time of isolation (*Figure 4D*). We conclude that *myog*[-/-] mutant and sib MuSCs have similar proliferative potentiality when assayed in culture.

To explore proliferation dynamics further, MPCs were pulsed with EdU at either 2, 3 or 4 days ex vivo after myofibre plating in growth medium (*Figure 4E*). To increase MPC yield for downstream analyses, 90–100 fibres were plated per well (*Figure 4E* and *Figure 4—figure supplement 1B*). As expected, *myog*[-/-] myofibres yielded more MPCs after 2 days in culture compared to sibs, and both genotypes produced mainly Desmin[+ve] muscle lineage cells (*Figure 4—figure supplement 1B,C*). An EdU pulse 2 days after plating revealed striking differences between *myog*[-/-] MPCs, 60% of which were EdU[+ve], compared to sib MPCs, only 20% of which incorporated EdU (*Figure 4E,F*). A day later (on day 3), the difference in proliferation had reversed; sib MPCs were almost 60% EdU[+ve], whereas incorporation into *myog*[-/-] MPCs was significantly reduced compared to both sibs on day 3 and *myog*[-/-] MPCs on day 2 (*Figure 4E,F*). This difference in S-phase labelling between *myog*[-/-] and sib MPCs persisted on day 4 (*Figure 4E,F*). Statistical analysis of overall EdU incorporation dynamics as Area Under Curve (AUC) revealed slightly higher proliferation rate of sib MPCs (*Figure 4E*). Although we have not analysed EdU incorporation at day 1 (when many MPCs are still on their

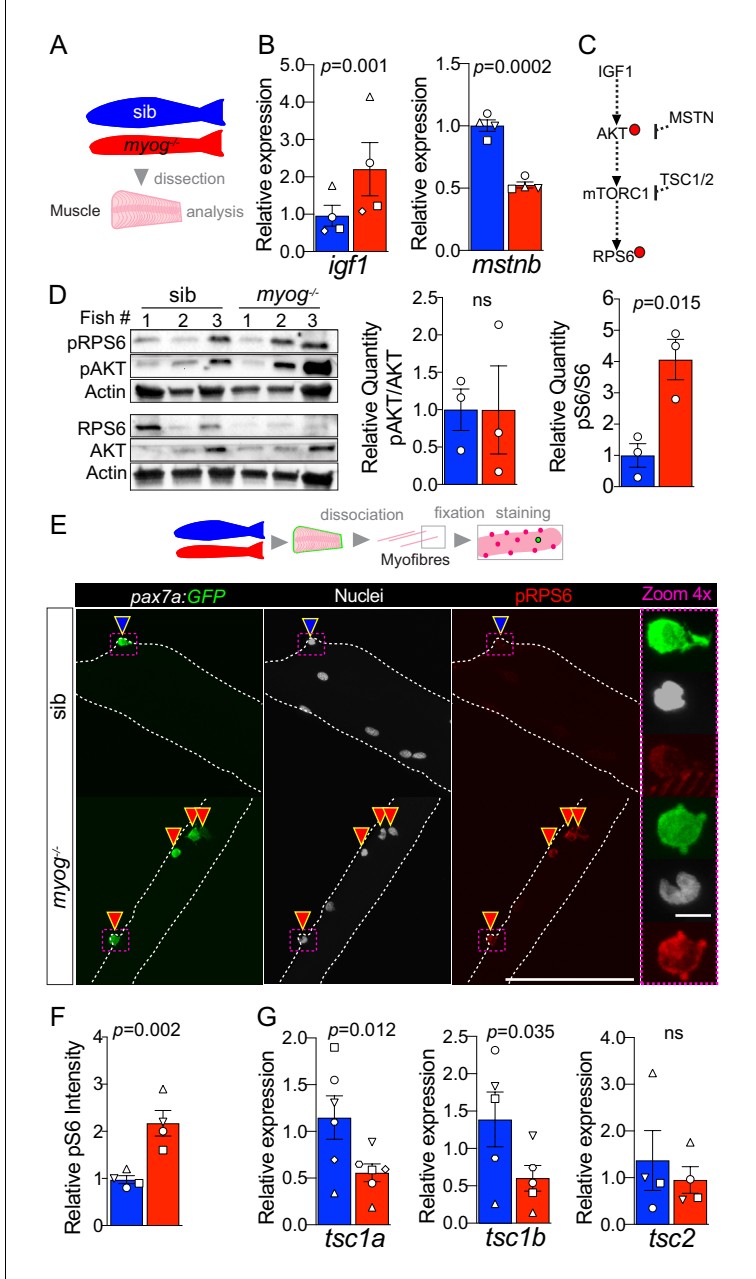

**Figure 3.** *Myog*[-/-] mutant muscle and MuSCs display enhanced mTORC1 signalling. (**A**) Schematic of adult trunk muscle processing for analysis. Colours identify sib (blue) or *myog*[-/-] (red) samples throughout the figure. (**B**) qPCR analysis shows upregulation of *igf1* and downregulation of *mstnb* in *myog*[-/-] muscle. Symbol shapes denote paired sib and *myog*[-/-] samples, n = 4 fish/genotype, paired *t*-test. (**C**) Summary of mTORC1 pathway with analysed components, positive (arrows) or negative (bars) effects are indicated. Dashed lines indicate other molecules involved; red dots represent phosphorylation events. (**D**) Western blot (left) and quantification (right) of phosphorylated/pan AKT (pAKT, Ser473) ratio and phosphorylated/pan RPS6 (pRPS6, Ser240/244). pRPS6 is increased in *myog*[-/-] muscle, n = 3 fish/genotype, unpaired *t*-test. Actin immunoreactivity was used to normalise pan and phospho signals to protein loading prior to calculation of sample-specific phospho/pan ratio. (**E**) Representative images of myofibre immunostained for GFP (green), pRPS6 (red) and nuclei (white) fixed freshy immediately after isolation. Arrowheads indicate MuSCs on sib (blue) or *myog*[-/-] (red) myofibres. Dashed magenta squares highlight 4x-magnified cells. Scale bar = 100 μm (=5 μm in 4x-zoom). (**F**) Quantification of pRPS6 intensity shows increase in *myog*[-/-] MuSCs. Symbol shapes denote paired sib and *myog*[-/-] samples, n = 4 fish/genotype, n = 20–30 MuSCs/fish, paired *t*-test. (**G**) qPCR analysis showing downregulation of *tsc1a* and *tsc1b* mRNAs but

*Figure 3 continued on next page*

*Figure 3 continued*

unaltered *tsc2* mRNA in *myog*[-/-] muscle. Symbol shapes denote paired sib and *myog*[-/-] samples, n = 4–6 fish/
genotype, paired *t*-test. All graphs report mean ± SEM.

The online version of this article includes the following figure supplement(s) for figure 3:

**Figure supplement 1.** *Myog*[-/-] mutant muscle displays unaltered *igfr1a* and *igfr1b* mRNA expression.

---

myofibres which, being often lost during staining, prevented analysis), these data clearly reveal that proliferation is not increased in mutant MPCs, simply activated earlier. We conclude that MPCs in *myog*[-/-] mutants are more readily driven into proliferation upon release from their in vivo environment.

We next explored MuSC dynamics by analysing MPCs mRNA levels at day 2 and day 3 of culture. Both *pax7a* and *pax7b* were more abundant, fourfold and threefold, respectively, in *myog*[-/-] MPCs at 2 days compared to sib (*Figure 4G*), consistent with the higher percentage of MuSCs observed (*Figure 4C*). Strikingly, while *pax7a* mRNA reduced with time, *pax7b* mRNA was maintained at similar level between day 2 and day 3 across genotypes, despite confirming overall increased level in *myog*[-/-] MPCs (*Figure 4G*). Moreover, *myod* mRNA was fifteenfold more abundant in *myog*[-/-] MPCs compared to sib, again suggesting an earlier entry into activation phase. Notably, higher *myod* expression in *myog*[-/-] MPCs lasted into day 3. In contrast, *myf5* mRNA level appeared similar in *myog*[-/-] and sib MPCs at both 2 and 3 days, although tending to decrease at the later timepoint in sibs (*Figure 4G*), perhaps reflecting their slower activation. The data suggest that *myog*[-/-] MPCs show faster activation and greater recovery from a stem-like state than MPCs from sibs.

The decreased proliferation but enhanced *myod* mRNA in cultured *myog*[-/-] compared to sib MPCs also suggested that the mutant cultures might be entering the terminal differentiation program despite the presence of growth medium, presumably due to their rapid proliferation resulting in the formation of confluent cell clusters that consequently begin myogenic terminal differentiation prematurely. We assessed onset of differentiation by analysing expression of *mef2d*, *mylpfa* (encoding a fast myosin light chain), *flnca* and *mybpc1*, crucial players in myogenesis, which revealed significantly higher levels of all the assessed mRNAs in *myog*[-/-] relative to sib MPCs on day 3, with an approximate upregulation of *mef2d* (twofold), *mylpfa* (fortyfold), *flnca* (fivefold) and *mybpc1* (thirtyfold) (*Figure 4H*). Among the four, only *mylpfa* showed accumulation in *myog*[-/-] MPCs already at day 2. In addition, mutant MPCs significantly accumulated *cdkn1a* (*p21*) mRNA (fourfold) between day 2 and day 3, in line with reduced proliferation and myogenic progression, whereas sib MPCs did not (*Figure 4—figure supplement 1D*). In contrast, level of *cdkn1ca* (*p57*), which promotes terminal myogenesis (*Osborn et al., 2011*), was unaltered in both *myog*[-/-] and sib MPCs (*Figure 4—figure supplement 1D*), highlighting the immature state of myogenic progression. However, despite the continued exposure to growth medium, immunostaining confirmed accumulation of sarcomeric myosin heavy chain (MyHC) and significantly higher differentiation index in *myog*[-/-] compared to sib MPCs at day 3, (*Figure 4I*). As reported previously (*Ganassi et al., 2018*), upon culture under differentiation conditions, sib MPCs underwent cell fusion forming visible multinucleated myotubes, whereas differentiated *myog*[-/-] MPCs remained mostly mononucleated (*Figure 4—figure supplement 1E*). Altogether, our data indicate that whereas Myog function is dispensable for MPC proliferation and terminal differentiation, its lack either prevents MuSCs achieving full quiescence or accelerates MuSC transition into the proliferation phase, suggesting that Myog contributes to maintaining the MuSC niche.

## Discussion

Here, we describe a novel function for the transcription factor Myog in regulating adult skeletal muscle growth rate and MuSC dynamics through four major findings. First, Myog influences MuSC number. Second, Myog is required for MuSCs to adopt their normal niche position. Third, Myog contributes to MuSC deep quiescence, regulating expression of genes involved in mTORC1 signalling. Lastly, Myog is required for proper myofibre growth and myonuclear accretion throughout life.

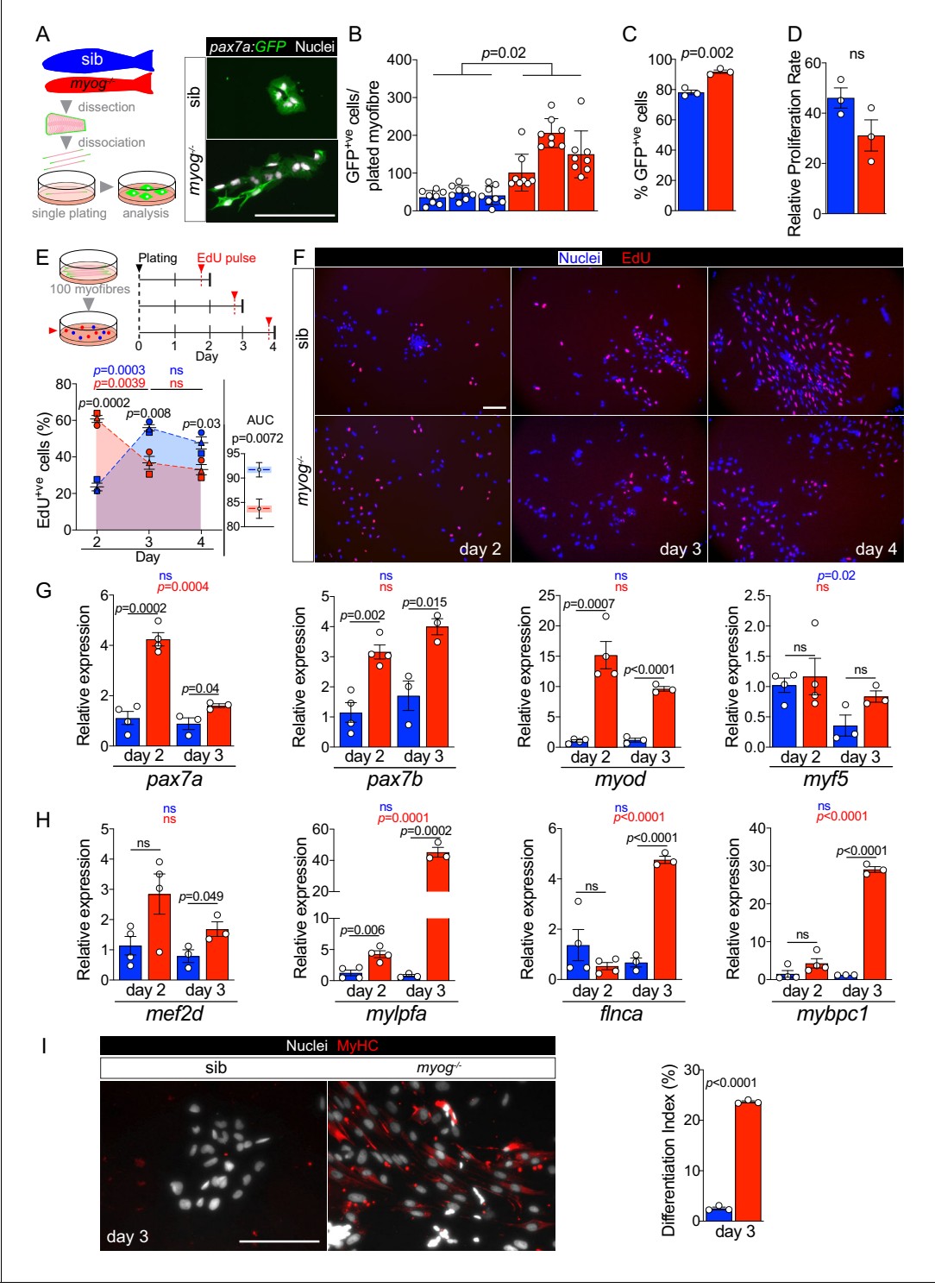

**Figure 4.** *Myog*<sup>-/-</sup> mutant MPCs exhibit faster entrance into proliferation phase. (A) Schematic of adult single myofibre culture and representative immunodetection of GFP (green) and nuclei (white) on cultured MPCs. Colours identify sib (blue) or *myog*<sup>-/-</sup> (red) samples throughout the figure. (B) Quantification of GFP<sup>+ve</sup> cells obtained by single myofibre culture revealed increased MPCs yield from *myog*<sup>-/-</sup> after 4 days in growth medium, n = 3 fish/genotype, n = 8 myofibres/fish. Unpaired *t*-test on average MPCs/fish/genotype. (C) Fraction of GFP<sup>+ve</sup> cells out of total number of cells obtained in B. n = 3 fish/genotype, unpaired *t*-test. (D) Relative proliferation rate of sib or *myog*<sup>-/-</sup> MPCs calculated as the ratio of the average number of GFP<sup>+ve</sup> MPCs/plated-myofibre obtained on day 4 in culture divided by the average number of GFP<sup>+ve</sup> MuSCs/myofibre, unpaired *t*-test. (E) Diagram of myofibres culture and EdU pulse regime (top). Quantification of fraction of EdU<sup>+ve</sup> MPCs at indicated time points (bottom left), symbol shapes denote MPCs obtained from paired sib and *myog*<sup>-/-</sup> samples, n = 3. Measurement of Area Under Curve (AUC, coloured area underneath dashed blue or red

*Figure 4 continued on next page*

*Figure 4 continued*

lines) to compare overall proliferation dynamics indicates reduction in *myog*[-/-] MPCs (bottom right), unpaired *t*-test. (F) Representative images of sib and *myog*[-/-] MPC numbers and EdU incorporation at indicated times in culture, showing detection of EdU (red) and nuclei (blue). (G) qPCR analysis showing expression dynamics of *pax7a*, *pax7b*, *myod* and *myf5* mRNAs in sib and *myog*[-/-] MPCs cultured for 2 (n = 4) or 3 days (n = 3), unpaired *t*-test. (H) qPCR analysis indicating upregulation of *mef2d*, *mylpfa*, *flnca* and *mybpc1* mRNAs in *myog*[-/-] MPCs cultured in growth medium for 2 (n = 4) or 3 days (n = 3), unpaired *t*-test. (I) Representative immunodetection of MyHC (red) and nuclei (white) on sib or *myog*[-/-] MPCs at 3 days in growth medium for 3 days (left). Extent of differentiation (Index (%), see Materials and methods) is higher in *myog*[-/-] than in sib MPCs (right). All graphs report mean ± SEM. Statistical significance within (coloured *p*) or between (black *p*) genotypes is indicated. Scale bars = 100 μm.

The online version of this article includes the following figure supplement(s) for figure 4:

**Figure supplement 1.** Characteristics of cultured MPCs and mRNA analysis.

## Myog controls MuSC number

We show that *myog*[-/-] mutation causes supernumerary MuSCs/MPCs, paralleled by increased *pax7a* and *pax7b* mRNA. In addition to the fivefold increase in MuSC/adult myofibre and 50% reduction in myofibre length reported here, we previously described that adult *myog*[-/-] mutants have unaltered body length and numbers of myofibre profiles/transverse body section roughly equal to their non-mutant siblings (*Ganassi et al., 2018*). Taken together, these data suggest that total number of Pax7[+] MuSCs is increased around 10-fold in adult *myog*[-/-] mutant myotome. In mouse, expression of Pax7 and Myog in MuSC appear to be mutually exclusive and controlled through reciprocal inhibition in in vitro studies (*Olguin et al., 2007*; *Riuzzi et al., 2014*). Persistent *Pax7* expression delays Myog accumulation in cultured myoblasts and ex vivo in MuSCs on myofibres, whereas silencing of *Myog* can result in retention of *Pax7* expression in differentiation (*Olguin et al., 2007*; *Zammit et al., 2006*). As Myog is required for adult myoblast fusion in vitro (*Ganassi et al., 2018*) and *mymk* and *mymx* mRNAs are reduced in adult *myog*[-/-] muscle, one must consider the possibility that some *pax7a:GFP*[+ve] mononucleate cells are not MuSCs but differentiated myocytes that retain the *pax7a* reporter and are unable to fuse. However, we detect Pax3/7 protein in MuSCs, showing that strong GFP accumulation is not just perdurance of earlier signal. Moreover, most *myog*[-/-] MuSCs enter into S-phase after 2 days ex vivo. Subsequently, *myog*[-/-] MPCs quickly upregulate the differentiation marker *mylpfa* fortyfold, suggesting that an insignificant fraction was previously terminally differentiated. In addition, *myog*[-/-] muscle expresses more *myf5*, a MPC marker. In post-natal rodent muscle, *Myf5* is expressed in MuSCs, not myofibres (*Beauchamp et al., 2000*). Like our finding, murine *Myog*-knockout also led to *Myf5* accumulation in neonatal limb muscle, even though one-third of the total nuclei are still MuSCs/MPCs in wild type neonates (*Cardasis and Cooper, 1975*; *Rawls et al., 1998*). Taken together, these data argue that the abundant *pax7a:GFP*[+ve] cells are MuSCs or MPCs.

We consider two possible hypotheses for the increased number of MuSC/MPCs in *myog*[-/-] mutants; differentiation failure or active accumulation. We do not favour the idea that absence of Myog prevents MPCs undergoing terminal differentiation leading to the accumulation of cells blocked in differentiation for three reasons. First, *myog*[-/-] MPCs readily undergo differentiation in culture and during myogenesis in the embryo (*Ganassi et al., 2018*). Second, although muscle is reduced in size, there is enough formed to support life and sarcomere length is unaffected, so terminal differentiation is fairly efficient without Myog during larval and adult growth and any required muscle repair. Third, both in fish and amniotes, MPCs ready to differentiate express high levels of Myod, which acts in a feedforward mechanism to trigger cell cycle exit and Myog expression (*Weintraub et al., 1991*; *Hinits et al., 2009*). We do not, however, observe increased levels of *myod* mRNA in *myog*[-/-] mutant muscle. Nevertheless, when *myog*[-/-] MPCs are placed in culture they dramatically upregulate *myod*, arguing that Myod upregulation is also a characteristic of adult zebrafish MPC myogenic progression. Active accumulation of MuSCs, on the other hand, is suggested by the increase in *pax7a*, *pax7b* and *myf5* mRNAs in *myog*[-/-] mutant muscle. Below, we raise the hypothesis that the abundance of myofibres with low numbers of nuclei may trigger MuSC accumulation as a homeostatic response to tissue insufficiency.

## Myog is required for MuSCs to adopt their normal niche position

In wild-type fish 21% of myofibres have no associated MuSCs, despite having around 95 myofibre nuclei. This finding strongly argues that, in order to grow or repair damage, MuSC/MPCs must migrate across the basal lamina between fibres, as occurs in developing rodent muscle (*Hughes and Blau, 1990*). We observe 0.92 MuSCs/wild-type myofibre, on average. If one assumes random MuSC distribution amongst myofibres, the Poisson distribution predicts that 40% of myofibres should have no associated-MuSCs. The lower proportion of myofibres lacking MuSCs suggests that MuSCs are not distributed randomly. Conversely, Poissonian distribution would predict that 37% of myofibres should have a single MuSC. The observed value of 70% strongly suggests that MuSCs actively disperse between myofibres. Nevertheless, the system is imperfect, as the 21% of myofibres lacking MuSCs shows. Despite the fivefold increase in MuSCs in $myog^{-/-}$ mutant, around 5% of myofibres still have no associated MuSC, which is far more than expected by the Poisson distribution (1.5%). Similarly, the 33% of $myog^{-/-}$ myofibres with only a single MuSC compares with an expected value of 6%. In contrast, $myog^{-/-}$ myofibres with over 10 MuSCs are over-represented, perhaps due to rare local regenerative events. In the absence of Myog, MuSCs tend to cluster on some myofibres, while also showing a tendency to be present at just one per myofibre more often than expected. These observations raise the possibility that myofibres in both wild-type and $myog^{-/-}$ mutant contain a single specific niche for MuSCs.

Most MuSCs associated with isolated myofibres in zebrafish are located near myofibre ends, which would be attached to the vertical myoseptum tendon-like structure in vivo. We hypothesise that the fibre end provides a special niche in which MuSCs accumulate, perhaps formed in association with the myotendinous junction (MTJ). Strikingly, the preferred location of MuSCs in wild-type myofibres is between 10 and 20 µm from the myofibre end, suggesting that MuSCs locate not on the MTJ surface itself, but on the immediately adjacent cylindrical sarcolemma, retaining a potential physical contact with the MTJ. Moreover, this suggested location of the MuSC niche may relate to evidence that addition of new nuclei occurs preferentially at the ends of myofibres during growth (*Kitiyakara and Angevine, 1963*; *Aziz and Goldspink, 1974*). Upon explant ex vivo, MuSCs activate and begin to migrate away from myofibre ends. In $myog^{-/-}$ mutants, the increased number of MuSCs are almost randomly distributed along the fibre, but the slightly higher abundance at myofibre ends might suggest that cells disperse once the MTJ niche is filled. However, when the fraction of myofibres in $myog^{-/-}$ mutants that have only a single MuSC were analysed separately, significant dispersal was still observed. An additional possibility is that loss of MTJ contact by the short myofibres in $myog^{-/-}$ mutants contributes to niche disruption. We conclude that Myog is required for assembly of the myofibre end MuSC:niche complex.

Mammalian MuSCs are generally not associated with myofibre ends, perhaps because myofibres are longer. Zebrafish myofibres are on average 1 mm long and rarely exceed 2 mm. In contrast, in the murine muscles most frequently analysed, those of the limbs, myofibres are up to 1 cm long. We suggest that the more numerous nuclei and larger number of MuSCs required for efficient growth and repair have allowed (or selected for) the MuSC niche to disperse along the myofibre.

## Myog contributes to MuSC deep quiescence

Numerous lines of evidence argue that MuSCs in $myog^{-/-}$ zebrafish fail to enter full quiescence. First, MuSCs are more numerous in $myog^{-/-}$ mutant adults. Second, they are often not in their normal niche. Third, they contain more pRPS6 upon acute isolation. Fourth, they activate and proliferate more rapidly upon ex vivo culture. Three hypotheses could explain the MuSC $myog^{-/-}$ phenotype. (1) Lack of Myog function in early development. However, MuSCs are not increased in mutant larvae (*Ganassi et al., 2018*) so it seems the defect worsens over the life course, suggesting a continuous need for Myog. (2) Lack of Myog within myofibre nuclei, which might lead to signals promoting MuSC proliferation (see below). (3) Myog could also function within MuSC themselves to control the balance between quiescence and activation. Whereas most literature describes *Myog* expression at the differentiation step following MuSC proliferation, various reports that noted Myog⁺ MuSCs support a Myog MuSC-specific function. An early study investigating differential expression of *Myod* and *Myog* in rat muscle observed rare but intense accumulation of Myog mRNA along myofibre edges, anatomically reminiscent of MuSC niche (*Hughes et al., 1993*). Notably, Myog expression predominates in oxidative myofibres also characterised by higher MuSC density (*Gibson and*

*Schultz, 1982*; *Hughes et al., 1993*). Rantanen et al. identified a population of dormant Myog[+] MuSCs which may contribute to muscle repair bypassing the proliferation step (*Rantanen et al., 1995*). Co-expression of Pax7 and Myog was observed in a satellite-cell study in transverse sections of both mouse soleus and EDL muscles (*Schultz et al., 2006*) and in rare cells in fish dermomyotome (*Devoto et al., 2006*). Likewise, immunohistochemistry on human resting muscle biopsies found *Myog* expression in MuSC but not in myonuclei (*Lindström et al., 2010*). More recently, the Blau group reported a low but detectable level of Myog protein in both resting murine MuSCs and in MPCs returning to quiescence after recovery from injury (*Porpiglia et al., 2017*). In addition, data-sets show that *Myog* mRNA is present in quiescent MuSCs and rapidly decreases significantly during the 3 hours of myofibre isolation (*Machado et al., 2017*). Interestingly, *Mrf4* expression in MuSC was downregulated in the same time frame, concomitant with exit from quiescence marked by the upregulation of *Myod*, demonstrating the presence of two well-known muscle 'differentiation-specific' markers in dormant stem cells. Together, such observations prompt re-evaluation of an intrinsic Myog function in quiescent MuSC.

## Myog is required for proper myofibre growth

MRFs including Myog are required in the adult to restrict murine myofibre size (*Moresi et al., 2010*; *Moretti et al., 2016*). We find that, in zebrafish, Myog is required both for myocyte fusion that per-mits increase in nuclear number and consequent growth to the normal fibre size. We also show that Myog limits myonuclear domain size both in juvenile and adult muscle. Conditional depletion of Myog in adult mouse muscle decreased myofibre cross-sectional area, although further morphomet-ric analysis of myofibre nuclear domains was not reported (*Meadows et al., 2008*). It therefore seems that, in addition to its role in promoting MPC differentiation and fusion, Myog restricts the volume of sarcoplasm supported by each myonucleus, both in denervated (*Moresi et al., 2010*) and innervated (this study) adult muscle. Both adult mouse and zebrafish *Myog* mutants show reduction in *Mrf4* mRNA (*Meadows et al., 2008*; *Venuti et al., 1995*; *Knapp et al., 2006*; *Rawls et al., 1998*; *Ganassi et al., 2018*). Mrf4 itself restricts murine myofibre growth (*Moretti et al., 2016*), although no myofibre size change was reported in the various *Mrf4* knockout mice (*Zhang et al., 1995*; *Patapoutian et al., 1995*). So Myog may either restrict myonuclear domain size directly or by regu-lating *Mrf4* expression.

Unlike fish, which grow to some extent throughout life, mice do not require MuSCs to maintain adult muscle size (*Keefe et al., 2015*). Nevertheless, during murine adult myogenesis myonuclear accretion from MuSC-derived MPC fusion is required for myofibre growth and regeneration, as in fish (*Pallafacchina et al., 2013*). Congruently, expression of fusogenic genes *myomixer* (*mymx*) and *myomaker* (*mymk*) is reduced in both *myog*[-/-] adult and embryonic muscles (*Ganassi et al., 2018*). Depletion of zebrafish Mymk leads to fewer myofibre nuclei in the surviving adults (*Shi et al., 2018*). Likewise, *Mymk*-KO in murine MuSCs at P0 led to 75% reduction of myonuclear number in extensor digitorum longus (EDL) myofibres measured at P28 (*Nikolaou et al., 2019*). Although severely reduced, accretion of myonuclei does occur in growing *myog*[-/-] zebrafish, thus confirming the persis-tence of a Myog-independent pathway to fusion (*Ganassi et al., 2018*), which appears to allow a fraction of MPCs to sustain some muscle growth.

Addition of myonuclei to growing muscle relies on interactive signals between MuSCs and myofi-bres. Fish lacking Myog have increased *igf1* and decreased *mstnb* expression. Both changes are pre-dicted to enhance muscle size (*Gao et al., 2016*; *Powell-Braxton et al., 1993*; *Coleman et al., 1995*; *Durieux et al., 2007*; *Trendelenburg et al., 2009*; *Zimmers et al., 2002*; *McPherron et al., 1997*), and both may be important in our *myog* mutants. As we observe in *myog*[-/-] fish, loss of Mstn increases MuSCs in amniotes (*McCroskery et al., 2003*; *Manceau et al., 2008*). MuSC contribution is, however, dispensable for myofibre hypertrophy in *Mstn* knockout mice, in which growth derives from increased myonuclear domain size. *Mstn* knockouts thus resemble both the MuSC increase and myofibre domain size increase in *myog* mutant fish. On the other hand, IGF1 has been shown to increase MPC proliferation and differentiation and thereby enhance myofibre growth, but without change in nuclear domain size (*Fiorotto et al., 2003*; *McCall et al., 1998*). *Igf1* increase might thus enhance the number of MuSCs in *myog* mutants.

IGF1 and Myostatin signalling also affect MuSC activation status in mice (*Cornelison and Wold, 1997*; *McCroskery et al., 2003*; *Zhang et al., 2010*; *Chakravarthy et al., 2001*) and other fish (*Garikipati and Rodgers, 2012*). Both can act through the mTORC1 pathway (*Trendelenburg et al.,*

*2009*; *Latres et al., 2005*). The IGF1 signal cascade phosphorylates AKT and activates mTORC1 leading to phosphorylation of RPS6 and eIF4EBPs, with downstream effects on both MuSCs and adult myofibres (*Schiaffino and Mammucari, 2011*). Loss of Mstn also activates the mTORC1/RPS6 pathway (*Trendelenburg et al., 2009*). Our data show enhanced RPS6 phosphorylation (pRPS6) both in $myog^{-/-}$ muscle and MuSC analysis ex vivo. Interestingly, mTORC1 controls the transition from deep $G_0$ quiescence to an intermediate pseudo-activated state defined as $G_{alert}$, in which MuSCs that accumulate pRPS6 are primed for activation compared to their quiescent counterparts (*Rodgers et al., 2014*; *Porpiglia et al., 2017*). Similar to zebrafish $myog^{-/-}$ MuSCs, 'alerted' mouse MuSCs display accelerated transition to proliferation ex-vivo compared to the quiescent population (*Rodgers et al., 2014*). As we detected no difference in Akt activation and unaltered levels of the receptors *igfr1a* and *igfr1b*, we deduce that increased phosphorylation of RPS6 is unlikely to result from enhanced upstream IGF1 cascade. However, $myog^{-/-}$ muscle showed significant downregulation of *tsc1a* and *tsc1b* mRNAs, that encode functional orthologues of mammalian TSC1, a mTORC1 inhibitor highly conserved from fly to human (*DiBella et al., 2009*; *Nobukini and Thomas, 2004*) that promotes stemness in various tissues (*Chen et al., 2008*; *Quan et al., 2013*). Mouse $Tsc^{-/-}$ MuSCs display enhanced pRPS6 and accelerated entry into proliferation in vivo and in vitro (*Rodgers et al., 2014*). Murine Myog can bind to conserved regions in *Tsc1* gene, whereas Myod does not, thus suggesting direct transcriptional regulation (Barbara Wold laboratory, Caltech; https://www.encodeproject.org/experiments/ENCSR000AID/) (*ENCODE Project Consortium, 2012*). These observations raise the hypothesis that loss of Myog leads to mTORC1 activation in MuSCs, exit from $G_0$ into $G_{alert}$ and subsequent increase in MuSC number. Taken together, our findings suggest that loss of Myog acts in adult animals to influence both MuSCs and muscle fibres, thus acting as a coordinator of tissue homeostasis.

# Materials and methods

## Key resources table

| Reagent type (species) or resource | Designation | Source or reference | Identifiers | Additional information |
|---|---|---|---|---|
| Genetic reagent (*D. rerio*) | $myog^{fh265}$, (AB) | Hughes/Moens Labs, *Hinits et al., 2011* | $myog^{fh265}$; RRID:ZFIN_ZDB-GENO-200128-7 | |
| Genetic reagent (*D. rerio*) | $myog^{kg125}$, (TL) | Hughes Lab, *Ganassi et al., 2018* | $myog^{kg125/+}$ (sib) $myog^{-/-}$ (mut); RRID:ZFIN_ZDB-GENO-200128-8 | |
| Genetic reagent (*D. rerio*) | $TgBAC(pax7a: GFP)^{t32239Tg}$, (AB) | Nüsslein-Volhard Lab, *Mahalwar et al., 2014* | pax7a:GFP; RRID:ZFIN_ZDB-GENO-170316-2 | |
| Genetic reagent (*D. rerio*) | $myog^{kg125}$;$TgBAC(pax7a:GFP)^{t32239Tg}$, (AB) | This paper | pax7a:GFP;$myog^{kg125}$ | Generated from outcrossing $myog^{kg125}$ onto pax7a:GFP |
| Antibody | Anti-phospho-S6 ribosomal protein (Ser 240/244) ( Rabbit monoclonal) | Cell Signalling | #5364; D68F8; RRID:AB_10694233 | (1:1000) IF (1:2000) WB |
| Antibody | Anti-S6 ribosomal protein (pan) (Mouse monoclonal) | Cell Signalling | #2317; 54D2; RRID:AB_2238583 | (1:1000) |
| Antibody | Anti-phospho-Akt (Ser473) (Rabbit monoclonal) | Cell Signalling | #4060; D9E; RRID:AB_2315049 | (1:1000) |
| Antibody | Anti-Akt (pan) (Mouse monoclonal) | Cell Signalling | #2920; 40D4; RRID:AB_1147620 | (1:1000) |
| Antibody | Anti-beta-Actin (Mouse monoclonal) | Sigma Aldrich | #A5316; AC-74; RRID:AB_476743 | (1:500) |

*Continued on next page*

*Continued*

| Reagent type (species) or resource | Designation | Source or reference | Identifiers | Additional information |
|---|---|---|---|---|
| Antibody | HRP goat-anti mouse IgG(H+L) (Goat polyclonal) | Sigma Aldrich | #AP308P; RRID:AB_92635 | (1:5000) |
| Antibody | HRP goat anti - abbit IgG (H+L) (Goat polyclonal) | Sigma Aldrich | #AP307P; RRID:AB_11212848 | (1:5000) |
| Antibody | Anti-Pax3/Pax7 DP312 (Mouse monoclonal) | Nipam Patel, UC Berkeley, USA | DP312 | (1:50) |
| Antibody | A4.1025 anti-Myosin heavy chain (Mouse monoclonal) | Hughes Lab, *Blagden et al., 1997* Developmental Studies Hybridoma Bank (DSHB) Merck | DSHB #A4.1025; Merck #05–716; RRID:AB_309930 | (1:5) |
| Antibody | MF20 anti-Myosin heavy chain (Mouse monoclonal) | Developmental Studies Hybridoma Bank (DSHB) | #MF20; RRID:AB_2147781 | (1:300) |
| Antibody | Anti-Desmin (Mouse monoclonal) | Sigma Aldrich | #D8281; RRID:AB_476910 | (1:100) |
| Antibody | Anti-GFP (Chicken polyclonal) | Abcam | #13970; RRID:AB_300798 | (1:400) |
| Antibody | Alexa Fluor 488 goat anti-chicken IgY (H+L) (Goat polyclonal) | Invitrogen | #A11039; RRID:AB_2534096 | (1:1000) |
| Antibody | Alexa Fluor 555 goat anti-rabbit IgG (H+L) (Rabbit polyclonal) | Invitrogen | #A27039; RRID:AB_2536100 | (1:1000) |
| Other | Hoechst 33342 nuclear stain | Thermo Fisher | #H3570 | (10 µg/ml) |
| Commercial assay or kit | Click-iT Plus EdU Cell Proliferation Kit for Imaging, Alexa Fluor 594 dye | Invitrogen | #C10639 | Used according to supplier instruction |
| Software, algorithm | Generalised linear models (GLS) | Adapted from: *Pinheiro et al., 2020*; *Zeileis, 2002*; *Ho et al., 2011* ; *Sekhon, 2011* | Growth mode analysis | |

## Zebrafish lines and maintenance

All lines used were reared at King's College London on a 14/10 hr light/dark cycle at 28.5°C, with staging and husbandry as described (*Westerfield, 2000*). $myog^{fh265}$ mutant allele were maintained on AB background and genotyped by sequencing as previously described (*Hinits et al., 2011*; *Roy et al., 2017*). $Myog^{kg125}$ ($myog^{-/-}$) were made by CRISPR/Cas9, genotyped as previously reported and maintained on TL background (*Ganassi et al., 2018*). $TgBAC(pax7a:GFP)^{t32239Tg}$, a generous gift from C. Nüsslein-Volhard (MPI Tübingen) (*Mahalwar et al., 2014*), and the newly generated $myog^{kg125};TgBAC(pax7a:GFP)^{t32239Tg}$ were maintained on AB background. All experiments were performed on zebrafish derived from F2 or later filial generation, in accordance with licences held under the UK Animals (Scientific Procedures) Act 1986 and later modifications and conforming to all relevant guidelines and regulations. Adult zebrafish measurement and analysis of weight, length, body mass index, and standard weight were performed as previously described (*Ganassi et al., 2018*).

## Isolation of zebrafish myofibres and culture of MuSCs from adult tissue

Isolation and culture of zebrafish adult muscle fibres was previously described (*Ganassi et al., 2018*). Zebrafish aged 8–9 months (adult) or 1 month (juvenile) were culled in high-dose tricaine (Sigma Aldrich), immersed for 5 min in 1% Virkon (3S Healthcare) diluted in $dH_2O$, washed in PBS for 5 min followed by 70% ethanol rinse, eviscerated and skinned. Fifteen-month-old adult were used for the

*myog^{fh265}* allele. For myofibre dissociation, trunk muscle was incubated in 0.2% Collagenase (C0130, Sigma Aldrich), 1% Penicillin/Streptomycin DMEM supplemented with 50 µg/ml gentamycin (Thermo Fisher) at 28.5°C for at least 2 hours. Single muscle myofibres were released by trituration using heat-polished glass pipettes and washed three times with DMEM Glutamax High Glucose (Gibco). Myofibres were then imaged for total length measure or plated on Matrigel (Invitrogen) coated 96- or 24-well plates and cultured in growth medium (20% Fetal Bovine Serum in 1% Penicillin/Strepto-mycin/DMEM GlutaMAX High Glucose supplemented with 10 µg/ml gentamycin). At indicated time points, MPCs were washed twice with PBS to remove plated myofibres, EdU pulsed (10 µM, Invitrogen Life Technologies) for 8 hr in fresh media and then fixed with 4% PFA for 15 min. For immunostaining myofibres were fixed in 4% PFA in PBS immediately after dissociation to reduce processing time and avoid MuSC activation.

## Myofibre morphometry and growth mode analysis

Absolute myofibre length was measured for viable (i.e. non-fixed) myofibres imaged immediately after isolation on Leica M stereo-microscope using x1.6 magnification. Measurement of Surface Area (SA) and number of nuclei was performed on fixed myofibres at x10 or x20 magnification using an Axiovert 200M microscope (Zeiss). Myofibre diameter was measured at two locations over total myofibre length. Morphometric calculation were carried out as described in *Brack et al., 2005*. Briefly, SA/unit length = myofibre segment length x $\pi$ x mean myofibre diameter; surface area domain size (SADS) = SA/myofibre nuclei in segment (Hoechst$^{+ve}$); Myofibre volume/unit length = myofibre segment length x $\pi$ x (radius)$^2$; myonuclear domain = myofibre segment length x $\pi$ x (radius)$^2$ /myonuclei (i.e. GFP$^{-ve}$ nuclei) in segment. MuSC (GFP$^{+ve}$ cell) distance to nearest myofibre-end was measured as schematised in *Figure 2—figure supplement 1G*. To graph frequency distribution, absolute nearest myofibre-end distances were normalised for myofibre length difference across genotype. Each myofibre was virtually divided in tenths where 1/10 and 5/10 corresponded to nearest segment to myofibre-end or segment at the myofibre half-length, respectively. Each tenth measured either 100 µm in sib or 50 µm in *myog$^{-/-}$*.

Generalised linear models (GLS) were used to explore growth assessing genotype differences in the relationship between SA and number of nuclei per 100 µm length (NoN), using the GLS function of NMLE package in R version 3.6.1 'Action of the Toes' (https://www.R-project.org/ *Pinheiro et al., 2020*). SA was log$_e$-transformed prior to analysis to generate a linear relationship with NoN, in a dataset pooling both juvenile and adult data. The dependent variable was set as NoN and the model included main effects of log$_e$(SA) and genotype, as well as an interaction term. The dataset was then split by genotype and separate GLS models were run to analyse the relationship between log$_e$(SA) and NoN for each genotype. *t*, *p*, and DF values were extracted for these models with the 'summary' function and were as follow:

- Interaction between genotype and log$_e$(SA): *t* = 7.332, *p*<0.0001, DF = 1164
- Siblings only, effect of log$_e$(SA): *t* = 11.448, *p*<0.0001, DF = 1,97
- Mutants only, effect of log$_e$(SA): *t* = 4.515, *p*<0.0001, DF = 1,67

Detailed steps of the GLS analysis are reported:

- Decision to pool age groups

Initial analyses showed no effect of age on the relationship between log$_e$(SA) and NoN, therefore different age groups were pooled to study the effect of mutation. Note that these initial analyses suffered from high confounding correlation between SA and age in siblings; however, visual analysis of the few datapoints where juvenile and adult SA values overlapped gave no indication that age was affecting the NoN directly.

- Decision to use Generalised Linear Model instead of General Linear Model

Significant heteroscedasticity was identified in simple linear models using R package lmtest (*Zeileis, 2002*), primarily because variance in NoN increased with log$_e$(SA). GLS models were used to weight explained variance by the value of the dependent variables, controlling for this heteroscedasticity. For each step of analysis, every combination of weighting (weighting by log$_e$(SA), genotype, both, or neither) was compared and the model with the lowest AIC was selected.

- Issue of Correlation between log$_e$(SA) and Genotype

In the final model, there was a significant correlation between explanatory variables $\log_e(SA)$ and genotype, which may have confounded interpretations. This correlation was removed by subsetting with the 'Matchit' and 'Matching' R packages, and qualitatively identical results were found by re-running the analysis using this smaller dataset (*Ho et al., 2011*; *Sekhon, 2011*).

## Protein extraction and western blot analysis

Western blot was performed as described (*Ganassi et al., 2014*; *Kelu et al., 2019*; *Kelu et al., 2020*). Briefly, dissected trunk muscle was submerged in lysis buffer (Tissue Extraction Reagent I [Invitrogen], Complete EDTA-free Protease Inhibitor Cocktail Tablets [Roche], 1 mM PMSF, 50 mM NaF, 1 mM $Na_3VO_4$ [Sigma Aldrich]) and triturated using TissueRuptor (Qiagen) followed by 5 min of sonication. Lysates were then pelleted by centrifugation, after which the supernatant protein extract was quantified, mixed with Laemmli Sample Buffer 4X (Bio-rad) complemented with 2-mercaptoethanol (Bio-rad) and heated at 95°C for 5 min, before subjecting to SDS-PAGE analysis. Protein extract equivalent to 50 μg was loaded per lane onto precast gradient gels (4–20% Bio-rad). Separated proteins were then transferred to nitrocellulose membranes, blocked in either 5% non-fat dry milk or 5% BSA in Tris-buffered saline and 0.1% Tween (TBST), incubated in primary and secondary antibodies at 4°C overnight and at room temperature for 2 hr, respectively. Primary antibodies used were: S6 ribosomal protein (pan) (1:1000; #2317; Cell Signaling), phospho-S6 ribosomal protein (Ser240/244) (1:2000; #5364; Cell Signaling), phospho-Akt (Ser473) 1:1000; #4060; Cell Signaling, Akt (pan) (1:1000; #2920; Cell Signaling), beta-Actin (1:500; #A5316; Sigma Aldrich). Secondary antibodies used were: HRP goat anti mouse IgG(H+L) (1:5000; #AP308P; Sigma Aldrich) and HRP goat anti rabbit IgG (H+L) (1:5000; #AP307P; Sigma Aldrich). Signal detection was performed using ChemiDoc Imaging System and analysed on Image Lab Software (Bio-rad). Phospho/pan (total) ratios were calculated following normalisation to membrane-matched and sample-matched Actin signals.

## Immunostaining on myofibres or cultured MPCs

For immunostaining, either myofibres or MPCs were permeabilised in PBS 0.5% Triton X-100 (PBST) for 15 min, blocked in Goat Serum 5% (Sigma Aldrich) in PBST and incubated with primary antibodies at indicated concentrations overnight in Goat Serum 2% in either PBST (0.1% TritonX) (myofibres) or PBS (MPCs). Primary antibodies used were against: phospho-S6 ribosomal protein (Ser240/244), (1:1000; #5364; Cell Signaling), GFP (13970 (1:400), Abcam), Myosin (A4.1025 (1:5), *Blagden et al., 1997*), MF20 (1:300, DSHB), Pax3/7 (DP312 (1:50), Nipam Patel, UC Berkeley, USA) and Desmin (D8281 (1:100), Sigma Aldrich). Samples were then washed three times in PBS prior to incubation with secondary antibodies in Goat Serum 2% in PBS. Secondary antibodies were Alexa Fluor 555 goat anti-rabbit IgG (H+L) (1:1000; #A27039; Invitrogen) and Alexa Fluor 488 goat anti-chicken IgY (H+L) (1:1000; #A11039; Invitrogen). Nuclei were counterstained with 10 μg/ml Hoechst 33342 (Thermo Fisher). EdU incorporation was revealed using a Click-iT EdU Imaging Kit (Invitrogen Life Technologies) as per manufacturer's instructions. Myofibres were imaged on LSM Exciter confocal microscope (Zeiss) equipped with x40 objective. MPCs were imaged at x10 or x20 using an Axiovert 200M microscope (Zeiss). At least three random fields were acquired in each of three technical replicates. Differentiation index was calculated as: nuclei in $MyHC^+$ myocytes x 100/total nuclei.

## RNA extraction, RT-PCR and qPCR

Genotyped $myog^{kg125/+}$ (sib) or $myog^{kg125}$ ($myog^{-/-}$) adult trunk muscles were triturated and processed for RNA extraction using RNA Purification Plus Kit (Norgen) following supplier's instructions. Total RNA was reverse transcribed using Superscript III reverse transcriptase (Invitrogen) following supplier's instructions. For MPCs, RNA extraction was performed at indicated time point using micro-RNA kit (Qiagen). Prior to collection, culture wells were washed three times with PBS to remove myofibres, followed by MPCs detachment using Accutase (Sigma Aldrich) for 10 min at 28.5° C. QPCR on technical triplicates for each sample was performed on 5 ng of relative RNA using Takyon Low ROX SYBR 2X MasterMix blue dTTP (Takyon) on a ViiAseven thermal cycler (Applied Biosystems). Ct values of all genes analysed were normalised to the geometrical mean of Ct values of three housekeeping genes (*actinb2, sep15* and *b2m*) and fold changes were calculated using ΔΔCt method (*Livak and Schmittgen, 2001*). Results are presented as mean value ± SEM of fold

changes from independent experiments as indicated. Used primers, purchased from Sigma-Aldrich (KiCqStart SYBR Green Primers Predesigned, Sigma Aldrich) are listed in *Supplementary file 1*.

## Statistical analyses

Quantitative analysis on images was performed with Fiji (NIH, www.Fiji.sc) and ZEN (2009 + 2012) software. Statistical analyses were performed using GraphPad (Prism 6 or 8) or Microsoft Excel (v16) for unpaired or paired two-tailed Student's *t*-test as indicated and one-way ANOVA followed by Holm-Sidak's posthoc (growth mode analysis) or Bonferroni (MuSC analysis). $\chi^2$ test was performed on raw data and used to assess difference between distributions.

## Acknowledgements

We are grateful to all members of the Hughes lab and Zammit lab for advice and to Bruno Correia da Silva and his staff for care of the fish. This work is supported by grants from the Medical Research Council to SMH (MRC Programme Grants G1001029 and MR/N021231/1) and PSZ (MR/P023215/1 and MR/S002472/1), and from Muscular Dystrophy UK (RA3/3052), Association Française contre les Myopathies (AFM17865) and the FSH Society (FSHS-82013–06 and FSHS-82017–05) to PSZ.

## Additional information

### Funding

| Funder | Grant reference number | Author |
|---|---|---|
| Medical Research Council | G1001029 | Simon M Hughes |
| Medical Research Council | MR/N021231/1 | Simon M Hughes |
| Medical Research Council | MR/P023215/1 | Peter S Zammit |
| Medical Research Council | MR/S002472/1 | Peter S Zammit |
| Muscular Dystrophy UK | RA3/3052 | Peter S Zammit |
| Association Française contre les Myopathies | AFM17865 | Peter S Zammit |
| FSH Society | FSHS-82013-06 | Peter S Zammit |
| FSH Society | FSHS-82017-05 | Peter S Zammit |

The funders had no role in study design, data collection and interpretation, or the decision to submit the work for publication.

### Author contributions

Massimo Ganassi, Conceptualization, Investigation, Methodology, Writing - original draft, Project administration; Sara Badodi, Investigation, Writing - review and editing; Kees Wanders, Data curation, Software; Peter S Zammit, Funding acquisition, Writing - review and editing; Simon M Hughes, Conceptualization, Formal analysis, Supervision, Funding acquisition, Writing - review and editing

### Author ORCIDs

Massimo Ganassi https://orcid.org/0000-0003-3163-9707
Sara Badodi https://orcid.org/0000-0002-8407-8336
Kees Wanders http://orcid.org/0000-0003-3209-9853
Peter S Zammit http://orcid.org/0000-0001-9562-3072
Simon M Hughes https://orcid.org/0000-0001-8227-9225

### Ethics

Animal experimentation: All experiments were performed on zebrafish derived from F2 or later filial generation, in accordance with licence held under the UK Animals (Scientific Procedures) Act 1986 and later modifications and conforming to all relevant guidelines and regulations.

Decision letter and Author response
Decision letter https://doi.org/10.7554/eLife.60445.sa1
Author response https://doi.org/10.7554/eLife.60445.sa2

## Additional files

### Supplementary files
- Supplementary file 1. List of qPCR primers used in the study.
- Transparent reporting form

### Data availability
All data generated or analysed during this study are included in the manuscript and supporting files.

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
