## [Decision Letter]

**Acceptance summary:**

This paper makes important contributions to our understanding of muscle homeostatic maintenance and muscle growth. In particular, using zebrafish mutants, the work shows that Myog is required for full quiescence of muscle stem cells and to locate to their proper 'niche' near the ends of myofibres. These insights may help to clarify causes of muscle decline in ageing and disease.

**Decision letter after peer review:**

Thank you for submitting your article "Myogenin is an Essential Regulator of Adult Myofibre Growth and Muscle Stem Cell Homeostasis" for consideration by *eLife*. Your article has been reviewed by three peer reviewers, one of whom is a member of our Board of Reviewing Editors, and the evaluation has been overseen by Didier Stainier as the Senior Editor. The following individual involved in review of your submission has agreed to reveal their identity: Clarissa Henry (Reviewer #2).

The reviewers have discussed the reviews with one another and the Reviewing Editor has drafted this decision to help you prepare a revised submission.

As the editors have judged that your manuscript is of interest, but as described below that additional analysis and quantifications of the experiments are required before it is published, we would like to draw your attention to changes in our revision policy that we have made in response to COVID-19 (https://elifesciences.org/articles/57162). First, because many researchers have temporarily lost access to the labs, we will give authors as much time as they need to submit revised manuscripts. We are also offering, if you choose, to post the manuscript to bioRxiv (if it is not already there) along with this decision letter and a formal designation that the manuscript is "in revision at *eLife*". Please let us know if you would like to pursue this option. (If your work is more suitable for medRxiv, you will need to post the preprint yourself, as the mechanisms for us to do so are still in development.)

Summary:

This manuscript uncovers an unexpected role for Myogenin in muscle stem/progenitor cells in adult zebrafish. Further analysis of a previously characterised mutant makes novel contributions to the field of muscle growth. The authors show that Myog helps keep MuSCs quiescent and provide mechanistic insights into how Myog controls MuSC activation. Intriguingly, their work suggests that Myog mutants have increased differentiation markers compared to wild-type siblings. They also offer new models for MuSC positioning within a fiber.

Essential revisions:

1) In their previous publication (Ganassi, 2018), the authors showed that the differentiation index of cultured adult myoblasts does not differ between WT and *myog^-/-^*, but in the current study myog mutant cells show a much higher degree of differentiation compared to their wild-type siblings. Please clarify why these findings differ.

2) The authors claim that the pax7a:GFP+VE cells represent bona fide MuSCs, which can only be determined by co-label of GFP and an anti Pax7 (or Anti Pax3+7) antibody. Although the authors do provide this co-label in one panel, they only show one cell per genotype. Please provide quantification of how often the GFP+VE cells are also positive for the anti-Pax3/7 antibody. Unless this co-label is extremely common in both genotypes, or confirmed in each experiment, the authors should soften their language about the GFP expressing cells being verified MuSCs.

3) Related to the previous point: The authors show that MPCs behave differently in culture and although they show increased pax7a and pax7b expression they also express higher levels of differentiation markers and enter terminal differentiation. This is puzzling and inconsistent with the reduced number of myonuclei and smaller myofibres that are seen in the mutant fish. Furthermore, it is not clear how the increased number of MuSCs per fibre is reached. A plausible explanation for both observations (fewer myonuclei/smaller fibres and more MPCs), is that these cells are myocytes that do not fuse efficiently. The authors raise this possibility in the Discussion, however, this should be better assessed and either excluded or supported.

4) The surface area domain size measurement in Figure 1 is a strange proxy for myonuclear domain. which is best thought of as a volume, as shown in Figure 2. However, Figure 2 omits all pax7:GFP+VE cells, some of which may have fused recently enough to retain their GFP label. Please replace the SADS calculation in Figure 1 with a volumetric calculation. This will be important for interpreting and comparing the two findings.

5) Culture of mononucleated MPCs from plated fibres was used to investigate whether lack of Myog enhanced MPC proliferation. The relative proliferation rates were not significantly different, however, EdU pulse experiments suggest that mutant MPC are more readily entering S-phase. Overall the authors suggest that lack of myog accelerates MuSC transition into the proliferation phase. At present this is not supported convincingly. Indeed, the data shows reduced proliferation and AUC (4E) in mutants. An additional EdU pulse at an earlier time after plating (day 1) should be included to potentially strengthen this idea. Alternatively the statement should be modified and toned down.

6) The authors want to assess whether there is an earlier onset of differentiation of MPCs in culture. However, they only show expression of *mef2d* and *mylpfa* at day 3 (Figure 4H) and day 2 should be included here as well. Overall the differentiation index of MPCs is increased, can they comment on whether the cells remain mononucleated.

7) In the Discussion the subsection “Myog is required for MuSCs to adopt their normal niche position” regarding the niche is very speculative and should be toned down/amended. In particular, the conclusion that Myog is required for assembly of the MTJ MuSC:niche complex is not well supported, there are no MTJ markers shown.

---

## [Author Response]

Essential revisions:1) In their previous publication (Ganassi, 2018), the authors showed that the differentiation index of cultured adult myoblasts does not differ between WT and myog^-/-^, but in the current study myog mutant cells show a much higher degree of differentiation compared to their wild-type siblings. Please clarify why these findings differ.

We think this result reflects the non-uniform distribution of MPCs in single fibre culture. In our previous analysis, similar numbers of MPCs from both mutant and sibling were replated and then triggered to differentiate by serum removal as a fair test of terminal differentiation capacity of myoblasts and yielded the result stated in Ganassi et al., 2018 of ‘no difference’. The reviewers are perspicacious to pick up the difference in the current analysis. When one plates cells in these 100 single fibre-derived cultures, MPC distribution is very non-random. The higher density of MPCs in mutant fibre cultures, and their rapid proliferation lead to a distribution in dense confluent clusters in the dish, which likely triggers their premature differentiation even in the presence of growth medium. In contrast, sibling cultures have fewer MuSCs to start with, slower proliferation and thus end up with fewer ‘colonies’ and at lower cell density. As a consequence, sibling MuSCs barely differentiate at 3 days in growth medium (Figure 4I). We believe this explains the difference and have now made this clear in the subsection “Adult *myog^-/-^* MuSC-derived myoblasts exhibit faster transition to proliferation”. Please note that in Figure 4—figure supplement 1E we show that the sibling cells in such cultures differentiate well in Differentiation Medium.

2) The authors claim that the pax7a:GFP+VE cells represent bona fide MuSCs, which can only be determined by co-label of GFP and an anti Pax7 (or Anti Pax3+7) antibody. Although the authors do provide this co-label in one panel, they only show one cell per genotype. Please provide quantification of how often the GFP+VE cells are also positive for the anti-Pax3/7 antibody. Unless this co-label is extremely common in both genotypes, or confirmed in each experiment, the authors should soften their language about the GFP expressing cells being verified MuSCs.

We take the reviewer’s point and have now performed such quantification. The answer is that 98.3%±1.7 (Sib) and 96%±1.9 (Mut; *myog^-/-^*) of GFP+ve cells are also co-labelled with DP312 (Pax3/Pax7) and are therefore real MuSCs. These data are now reported in Figure 2D and described in the subsection “Adult *myog^-/-^* myofibres have increased number of MuSCs”. An important additional point is that many GFP+ve cells go on to become EdU positive, and so are not myocytes, which are defined as having undergone irreversible cell cycle exit.

3) Related to the previous point: The authors show that MPCs behave differently in culture and although they show increased pax7a and pax7b expression they also express higher levels of differentiation markers and enter terminal differentiation. This is puzzling and inconsistent with the reduced number of myonuclei and smaller myofibres that are seen in the mutant fish. Furthermore, it is not clear how the increased number of MuSCs per fibre is reached. A plausible explanation for both observations (fewer myonuclei/smaller fibres and more MPCs), is that these cells are myocytes that do not fuse efficiently. The authors raise this possibility in the Discussion, however, this should be better assessed and either excluded or supported.

We suspect that mutant MuSCs are semi-quiescent (‘alerted’) in vivo, as indicated by accumulation of phosphorylated RPS6 in whole muscle analysis (Figure 3D), but become fully activated and rapidly enter the high proliferation phase ex vivo in culture, much faster than sib MuSCs. Our ex vivo DP312 staining on fibres fixed immediately upon dissection (Figure 2D) now confirms that virtually all of GFP+ve cells are MuSCs, thus not myocytes, with robust accumulation of RPS6 (Figure 3E). This, along with the high percentage of cells incorporating EdU at day 2 in proliferation medium, excludes the possibility that most mutant GFP+ve cells in vivo are unfused myocytes. Moreover, mutant MuSCs, similarly to Sib MuSCs, continue to proliferate on days 3 and 4 in culture, confirming a substantial proportion of myoblasts in both genotypes.

4) The surface area domain size measurement in Figure 1 is a strange proxy for myonuclear domain. which is best thought of as a volume, as shown in Figure 2. However, Figure 2 omits all pax7:GFP+VE cells, some of which may have fused recently enough to retain their GFP label. Please replace the SADS calculation in Figure 1 with a volumetric calculation. This will be important for interpreting and comparing the two findings.

We have now replaced SADS with the volume nuclear domain measure (Figure 1J), and relegated the SADS graph to supplementary Figure 1—figure supplement 1F. We call it ‘Nuclear Domain’, not ‘Myonuclear Domain’ to make it clear that it is calculated including MuSC nuclei.

Regarding the important recent-fusion suggestion, we do not believe this is confounding our results. We previously showed by FRAP (Bajanca et al., 2015) and by watching fusion in timelapse confocal microscopy (Pipalia et al., 2016) that cytoplasmic GFP diffuses rapidly (within seconds) throughout muscle fibre cytoplasm upon fusion. Due to the very high dilution factor, this effectively makes GFP from fused MPCs invisible without ultra-sensitive analysis (see videos in Pipalia et al., 2016). We have now made this logic clear in the subsection “Adult *myog^-/-^* myofibres have increased number of MuSCs”.

5) Culture of mononucleated MPCs from plated fibres was used to investigate whether lack of Myog enhanced MPC proliferation. The relative proliferation rates were not significantly different, however, EdU pulse experiments suggest that mutant MPC are more readily entering S-phase. Overall the authors suggest that lack of myog accelerates MuSC transition into the proliferation phase. At present this is not supported convincingly. Indeed, the data shows reduced proliferation and AUC (4E) in mutants. An additional EdU pulse at an earlier time after plating (day 1) should be included to potentially strengthen this idea. Alternatively the statement should be modified and toned down.

We may have confused readers by including the AUC analysis because, as the reviewers point out, such analysis may not reflect in detail the proliferation dynamics (i.e. is not the ‘whole’ proliferation curve). The main point we wanted to make is that *myog^-/-^* cells do not appear to proliferate *more* (i.e. this is not a terminal differentiation defect). The EdU data are very clear that there are almost threefold more EdU+ cells at day 2 in mutants compared to sibs, making a strong argument that entry into S-phase is temporally advanced in mutants compared to sibs. Combining this result with the alerted/pre-activated state revealed in Figure 3, the observations that sib cells reach a highly proliferative stage by day 3 (Figure 4E), and that differentiation is not reduced in mutants (Figure 4H, I) clearly shows that mutant MuSCs are entering the highly proliferative phase faster than those from sibs.

A practical problem is that on day 1 the isolated fibres do not yet bind Matrigel strongly enough to permit efficient MPC migration onto the dish surface, making the EdU analysis at day 1 technically difficult; too many fibres and associated MPCs get lost during labelling and staining. This is why we cannot include day 1 data, and is something commonly observed in single fibre cultures from mouse. We have now modified the subsection “Adult *myog^-/-^* MuSC-derived myoblasts exhibit faster transition to proliferation”, to make our argument more clear.

6) The authors want to assess whether there is an earlier onset of differentiation of MPCs in culture. However, they only show expression of mef2d and mylpfa at day 3 (Figure 4H) and day 2 should be included here as well. Overall the differentiation index of MPCs is increased, can they comment on whether the cells remain mononucleated.

We have now added further data to address both these points. Firstly, we have included an analysis of *mef2d* and *mylpfa* at day 2, as requested. Secondly, we have analysed other differentiation markers, including *cdkn1a* (p21), *cdkn1c* (p57), *flnca* (filamin C) and *mybpc1* (Myosin binding protein C1) at both day 2 and day 3. We report the new results in the subsection “Adult *myog^-/-^* MuSC-derived myoblasts exhibit faster transition to proliferation”, and Figure 4H and Figure 4—figure supplement 1D. Thirdly, we now include images showing that, after 5 days in differentiation condition (reduced serum), both sib and mutant MPCs differentiate but mutant myocytes remain largely mononucleated (Figure 4—figure supplement 1E), confirming our previous report (Ganassi et al., 2018). We have now made this clear in the aforementioned subsection.

7) In the Discussion the subsection “Myog is required for MuSCs to adopt their normal niche position” regarding the niche is very speculative and should be toned down/amended. In particular, the conclusion that Myog is required for assembly of the MTJ MuSC:niche complex is not well supported, there are no MTJ markers shown.

We have modified this discussion throughout the Discussion section to make it clear that we hypothesise MTJ involvement. The reviewers are correct that we have only shown that the immediately MTJ-adjacent fibre end forms the niche.